# A Deep Learning Framework for Early Parkinson’s Disease Detection: Leveraging Spiral and Wave Handwriting Tasks with EfficientNetV2-S

**DOI:** 10.3390/diagnostics15212795

**Published:** 2025-11-04

**Authors:** Ayesha Razaq, Shabana Ramzan, Sohail Jabbar, Muhammad Munwar Iqbal, Muhammad Asif Habib, Umar Raza

**Affiliations:** 1Department of Computer Science and IT, Government Sadiq College Women University, Bahawalpur 63100, Pakistan; razaqayesha99@gmail.com (A.R.); shabana@gscwu.edu.pk (S.R.); 2College of Computer and Information Sciences, Imam Mohammad Ibn Saud Islamic University (IMSIU), Riyadh 11432, Saudi Arabia; maabid@imamu.edu.sa; 3Department of Computer Science, University of Engineering and Technology, Taxila 47050, Pakistan; munwar.iq@uettaxila.edu.pk; 4Department of Engineering, Manchester Metropolitan University, Manchester M1 5GD, UK; u.raza@mmu.ac.uk

**Keywords:** Parkinson’s disease, handwriting analysis, EfficientNetV2-S, histogram equalization, Canny edge detection, deep learning

## Abstract

**Background:** Early detection of Parkinson’s disease (PD) is vital for improving patient outcomes, yet traditional diagnostic methods often depend on subjective clinical evaluations. **Methods:** This study proposes a novel deep learning framework for PD detection based on spiral and wave handwriting patterns from the PaHaW dataset. A comprehensive preprocessing pipeline is implemented, integrating histogram equalization and Canny edge detection. The processed spiral and wave images are evaluated independently using a fine-tuned EfficientNetV2-S architecture for binary classification. In addition to the EfficientNetV2-S experiments, a baseline Convolutional Neural Network (CNN) model is implemented separately for the spiral and wave handwriting images. The proposed model is further assessed using a 5-fold cross-validation strategy to ensure robustness and generalizability. **Results:** The models achieved validation accuracies of 98.68% on the spiral dataset and 98.10% on the wave dataset, with high Receiver Operating Characteristic–Area Under the Curve (ROC–AUC) scores, indicating robust discrimination between healthy and PD subjects. Analysis of the confusion matrix and classification results confirmed consistent sensitivity and specificity across the dataset. The 5-fold cross-validation yielded a standard deviation of ±0.0109. **Conclusions:** These results highlight the strong potential of handwriting analysis for early PD detection.

## 1. Introduction

PD is a chronic and progressive neurodegenerative disorder primarily characterized by the degeneration of dopaminergic neurons in the substantia nigra. This neuronal loss disrupts motor control, manifesting in hallmark symptoms such as resting tremor, muscular rigidity, bradykinesia, and postural instability [1]. Current evidence shows that PD develops through a multifactorial process involving both genetic susceptibility and environmental factors, with pathological changes beginning years before a clinical diagnosis becomes possible [2]. Given its progressive nature, the early and accurate detection of PD is essential, as timely therapeutic intervention can help slow disease progression, alleviate symptoms, and significantly enhance patients’ quality of life. Traditional diagnosis of PD largely depends on clinical assessment of motor symptoms, commonly quantified using standardized tools such as the Unified Parkinson’s Disease Rating Scale (UPDRS) [3]. Advanced neuroimaging techniques like Magnetic Resonance Imaging (MRI) can aid diagnosis, but the development of automated systems using such data remains challenging [4]. The clinical relevance of this disease lies in the gradual loss of neuromuscular control, which is often reflected in subtle irregularities in drawing patterns. These include distortions, tremor-induced oscillations, and variations in stroke dynamics, which may appear well before overt clinical symptoms are detectable through conventional assessments [5]. Automated PD diagnosis can be performed using computer vision techniques applied to handwriting exams from a relatively large patient dataset. The use of image processing and machine learning (ML) methods to analyze handwriting, extract quantitative features, and measure tremor severity has shown encouraging potential for early detection [6]. Deep learning approaches using handwriting patterns have demonstrated remarkable efficacy, with optimized architectures achieving remarkable diagnostic accuracies [7]. Handwriting features such as pressure, velocity, and stroke dimensions can reliably distinguish Parkinson’s patients from healthy controls (HC) [8]. Dynamic handwriting analysis is another useful method for identifying early signs of PD [9,10].

Deep learning has significantly advanced handwriting-based PD analysis with Convolutional Neural Networks (CNNs). The key advantage of machine learning and deep learning (DL) is their versatility, as these techniques are now widely applied across healthcare [11,12,13,14,15], networking [16,17], and numerous other domains, demonstrating their capacity to solve diverse and complex problems. A broader review discusses methods that incorporate handwriting data, including motion and pressure signals captured with digital tablets, combined with advanced computational techniques, highlighting their promise for early detection [18]. Handwriting analysis, particularly through spiral and wave drawing tasks, has gained attention as a cost-effective approach for detecting motor impairments associated with early-stage PD [19]. Handwriting tasks involving static images of spirals or simple shapes and continuous sentence writing to capture temporal changes better reflect motor symptoms in a natural writing context. Digital spiral drawing tasks have proven effective in distinguishing Parkinson’s patients from healthy individuals, demonstrating their value as a practical assessment method and their potential for use in future diagnostic systems [20]. Transfer learning with pretrained models such as VGG, ResNet, and EfficientNet has improved accuracy by utilizing robust, generalizable features while requiring minimal task-specific tuning [21,22,23]. For instance, a recent study utilizing the Adam-EfficientNetB3 model on the UCI spiral and wave dataset achieved high accuracy, demonstrating the particular efficacy of the EfficientNet architecture for this task [24]. Hybrid and attention-driven models, including Swin Transformers enhanced with coordinate attention, leverage long-range dependencies in handwriting patterns to achieve greater robustness and accuracy in PD detection [25].

Although numerous studies have explored handwriting-based detection of PD, a significant research gap remains in the underutilization of advanced preprocessing to create enriched input representations that maximize feature saliency for deep learning models. Most prior works have either relied on handcrafted features or concentrated on single-task modalities such as spiral drawings. While these methods show promise, they often face challenges such as limited generalization, dependency on manually engineered features, and insufficient utilization of multimodal handwriting patterns. Moreover, many deep learning pipelines process raw grayscale handwriting data without incorporating preprocessing strategies that enhance feature saliency. To address this gap, we develop a robust and comprehensive framework that incorporates multiple advanced preprocessing stages, including grayscale conversion, histogram equalization, and edge detection. These processed outputs are merged into a synthetic RGB representation to retain contrast, structural details, and edge information. This novel multi-channel fusion technique, a core contribution of this work, provides an enhanced input that is then used to fine-tune the EfficientNetV2-S backbone for accurate classification of handwriting samples into PD and healthy categories. The framework is designed to prioritize both high diagnostic accuracy and interpretability. EfficientNetV2-S is a variant in the EfficientNetV2 family. The EfficientNetV2 architecture introduces advanced fusion mechanisms, along with a progressive learning strategy that dynamically adjusts data augmentation techniques and input image resolutions throughout the training process. This design enables the network to significantly reduce training times, achieving speeds up to eleven times faster while maintaining or even improving classification accuracy on benchmark datasets such as ImageNet and CIFAR compared to earlier architectures [26].

### 1.1. Study Contributions

In this study, we make the following contributions:We propose a synthetic channel fusion technique that integrates grayscale images, histogram-equalized images, and Canny edge maps into a pseudo-RGB representation. This improves visual discriminability and enables the model to exploit contrast and edge-based cues often overlooked in conventional pipelines.We fine-tune the EfficientNetV2-S architecture, well-known for its balance between representational power and computational efficiency, for the classification of spiral and wave handwriting tasks. Independent models are trained for each modality, with cross-validation ensuring robustness.Performance is assessed using multiple metrics, including accuracy, precision, recall, F1-score, ROC–AUC, and confusion matrix analysis, providing a balanced view of both sensitivity and specificity.

Through these contributions, this study demonstrates that preprocessing-enhanced, explainable deep learning pipelines can substantially improve handwriting-based PD detection.

### 1.2. Research Hypotheses

This study is guided by the following testable hypotheses to thoroughly assess the performance of the proposed EfficientNetV2-S-based framework:The use of both spiral and wave handwriting tasks provides complementary information that enhances the detection of PD.Advanced preprocessing techniques that enhance handwriting features, such as histogram equalization, edge detection, and synthetic RGB channel fusion, will improve the discriminative capacity of deep learning models compared to raw grayscale handwriting inputs.Fine-tuning the EfficientNetV2-S architecture on preprocessed handwriting images achieves good accuracy and robustness compared to traditional handcrafted feature-based approaches.

The rest of this paper is organized as follows. Section 2 reviews related works, emphasizing recent advances in handwriting-based PD detection and deep learning methodologies. Section 3 details the proposed framework, covering the dataset, preprocessing pipeline, model architecture, training configuration, and evaluation metrics. Section 4 presents the experimental results, compares them with existing approaches, and discusses their implications and potential limitations. Section 5 concludes this study and outlines directions for future research.

## 2. Related Works

Early detection of Parkinson’s disease (PD) has become a major focus in recent research, with handwriting analysis proving to be a valuable diagnostic tool. Traditional methods primarily depend on manually engineered features, including pen velocity, acceleration, stroke pressure, and tremor frequency derived from spiral and wave drawing tasks. Although these approaches have shown promise in clinical settings, they often require extensive feature engineering and struggle to generalize effectively across different patients and varying recording conditions.

The authors of [27] investigated the dynamic handwriting characteristics of spiral drawing tasks and reported that conventional metrics yielded moderate classification accuracy, with Area Under the Curve (AUC) values ranging from 0.67 to 0.79. Their results underscore the importance of angular and reversal-based movement features as highly sensitive indicators for detecting PD. The authors of [28] proposed a dual-CNN framework for PD detection by analyzing both spiral and wave handwriting patterns. By integrating predictions using an ensemble voting classifier, they achieved an overall accuracy of 93.3%. The study in [29] presented a customized version of the SqueezeNet architecture specifically designed to identify PD based on handwriting characteristics. The proposed model achieved 90% accuracy, demonstrating its potential for predicting neurological disorders through the analysis of handwriting samples.

The authors of [25] proposed a Swin Transformer (CAS Transformer) for PD detection using handwriting analysis. Leveraging CycleGAN-based data augmentation and a tailored attention mechanism, the model attained an accuracy of 92.68%. The authors of [30] developed an automated framework for early PD detection by analyzing hand-drawn spiral and wave patterns using deep learning architectures, specifically Inception V3 and Xception, trained on more than 4,500 augmented images, achieving 97% accuracy.

The authors of [31] analyzed spiral and wave handwriting samples using a CNN-based model and reported that the VGG-19 model achieved an accuracy of 91.33% for distinguishing PD patients from HCs. The study demonstrated handwriting patterns as promising indicators for early PD detection while noting that reliance on specific drawing tasks may constrain the generalizability of the findings. The authors of [32] used sequences of sentences from 24 Parkinson’s patients and 24 HCs to analyze how handwriting evolves. They parameterized the images using domain-knowledge features and applied exhaustive feature selection with an SVM classifier, obtaining an overall accuracy of 91.67%. The study in [33] proposed a PD detection system using online handwriting, incorporating Beta-stroke segmentation, feature extraction with the Beta-elliptical method, a fuzzy perceptual detector, and classification via bidirectional LSTM. Evaluations on both the handwriting dataset and the PaHaW dataset demonstrated strong accuracy. The authors of [21] evaluated six pre-trained models (VGG16, VGG19, ResNet variants, and ViT) on a dataset of spiral and wave handwriting. They augmented the dataset with PixMix and AugMix techniques and applied a cosine annealing scheduler during fine-tuning, achieving 96.67% accuracy using the VGG19 model. The authors of [34] analyzed the diagnosis of PD using the VGG19 model with an attention mechanism, achieving 97.5% accuracy.

By evaluating the impact of Canny edge detection and Hessian filtering (HF) as preprocessing steps on various machine learning models, such as logistic regression, decision trees, random forests, and support vector machines, in distinguishing individuals at risk of PD, [35] found that while dataset augmentation improved model performance, applying edge-detection preprocessing actually degraded accuracy for most classifiers. The authors of [36] explored a hybrid architecture that integrates MobileNet with LSTM layers to analyze handwritten spiral drawings for PD detection. This approach demonstrated a notable performance gain, achieving approximately 87% accuracy on a public Parkinson’s Drawings Dataset. In [37], the authors introduced a CNN-based ACC-Net model to enhance feature extraction from PD spiral and wave drawings with an attention mechanism, achieving an accuracy of 96.5%. The authors of [38] proposed a sophisticated hybrid architecture (ILN-GNet), combining improved LinkNet and GhostNet for PD detection from handwriting. Their framework utilized a modified Wiener filter and multi-type feature fusion, achieving a high accuracy of 98.2% and outperforming several baseline models, including EfficientNet.

Table 1 summarizes the methodologies and comparative results from the related literature on PD detection using handwriting and drawing analysis. Despite notable progress with CNNs, Transformers, and ensemble models, few studies have explored dual-input frameworks that integrate both grayscale/RGB and edge-enhanced representations within a unified architecture. This research gap provided the motivation for our current work, which introduces an EfficientNetV2-S backbone framework that incorporates multiple advanced preprocessing stages, including grayscale conversion, histogram equalization, and edge detection, for accurate classification of handwriting samples into PD and healthy categories.

### Research Gaps

Although handwriting-based approaches for PD detection have achieved significant progress, several key research gaps remain. Early studies predominantly relied on kinematic- and pressure-based handcrafted features extracted from handwriting tasks. While these features provide valuable insights, they often lack robustness and fail to generalize across diverse datasets. Many deep learning pipelines use raw or grayscale handwriting images without additional preprocessing. Techniques such as contrast enhancement, Canny edge detection, or Laplacian filtering are underutilized, which may cause subtle but important motor irregularities to be missed. Despite the strong performance of CNN- and Transformer-based architectures, the interpretability of these models remains limited. Few studies incorporate explainability techniques such as Grad-CAM, saliency maps, or attention visualization to provide clinicians with actionable insights. Several studies rely on small, imbalanced datasets and lack rigorous validation strategies, raising concerns about model overfitting and real-world reliability. This research aims to address the identified gaps by introducing an EfficientNetV2-S-based framework that incorporates multiple advanced preprocessing stages, including grayscale conversion, histogram equalization, and edge detection, to enhance feature representation and enable accurate classification of handwriting samples into PD and healthy categories.

## 3. Methods

This section outlines the proposed framework for handwriting-based PD detection. The pipeline consists of dataset acquisition, data preprocessing, data augmentation, model training with EfficientNetV2-S, and comprehensive evaluation supported by explainability techniques. An overview of the complete workflow is shown in the methodology diagram in Figure 1.

### 3.1. Data Collection

The primary dataset employed in this study is Kaggle’s publicly available Parkinson’s Augmented Handwriting Dataset [39]. This dataset was specifically curated to facilitate research on the early detection of PD through handwriting analysis. To enhance diversity and mitigate overfitting, the dataset incorporates augmented variations of the original handwriting samples. It consists of spiral and wave handwriting images categorized into two classes: Parkinson’s and healthy. For experimental consistency, we divided the dataset into training, validation, and test sets. The spiral and wave datasets were balanced, each containing a total of 510 images, with 255 images per class. Specifically, for each task, the dataset was partitioned into a training set of 356 images (178 Parkinson’s, 178 healthy), a validation set of 76 images (38 Parkinson’s, 38 healthy), and a test set of 78 images (39 Parkinson’s, 39 healthy). The dataset size employed in this study is consistent with the scale of datasets used in other deep learning studies for PD detection from handwriting [27,31,32], providing sufficient data points for training. This ensures that sufficient data are available for model training, hyperparameter optimization, and independent evaluation. Furthermore, during training, we applied data augmentation techniques, including random rotation, shift, zoom, shear transformation, and horizontal flip. These augmentations complemented the pre-existing augmented samples, further enriching the variability of the dataset and enhancing the generalization capability of the model. Figure 2 presents random sample images of spiral and wave handwriting from the training set.

### 3.2. Data Preprocessing

To ensure that handwriting images contained enhanced discriminative features for classification, a structured preprocessing pipeline was applied. The pipeline transformed raw handwriting inputs into enriched synthetic representations by emphasizing intensity, contrast, and structural information. Original images were converted from RGB to grayscale to reduce computational complexity while preserving stroke intensity, as shown in Equation (Equation 1):(1)$$I_{\mathrm{gray}}\left(x,y\right)=0.299\cdot R\left(x,y\right)+0.587\cdot G\left(x,y\right)+0.114\cdot B\left(x,y\right)$$

This compresses the three-channel input into a single channel, retaining luminance information sufficient for handwriting analysis. To enhance local contrast and make faint handwriting strokes more visible, histogram equalization redistributes intensity values across the available range, as shown in Equation (Equation 2):(2)$$s_{k}=\frac{\left(L-1\right)}{MN}\sum_{j=0}^{k}n_{j}$$
where *L* is the number of possible intensity levels, *M* × *N* is the image size, and $n_{j}$ is the count of pixels with intensity *j*. This step improves stroke clarity by enhancing contrast in darker or uneven regions. To emphasize structural features such as stroke boundaries and tremor-induced irregularities, the Canny edge detection algorithm is employed with fixed thresholds (threshold1 = 100, threshold2 = 200) and a 5 × 5 Gaussian kernel for optimal noise reduction while preserving relevant edge information. It involves Gaussian smoothing, gradient computation, non-maximum suppression, and double thresholding, as shown in Equation (Equation 3). Edges correspond to pixels where the gradient magnitude exceeds predefined thresholds, highlighting abrupt intensity changes along handwriting contours.(3)$$G\left(x,y\right)=\sqrt{{\left(\frac{\partial \;I}{\partial \;x}\right)}^{2}+{\left(\frac{\partial \;I}{\partial \;y}\right)}^{2}}$$

Equation (Equation 4) shows that the three processed images are combined into a synthetic RGB image, enabling the model to leverage complementary features: grayscale image (intensity), histogram-equalized image (contrast), and edge-detected image (structural details).(4)$$I_{\mathrm{fused}}\left(x,y\right)=\left[I_{\mathrm{gray}}\left(x,y\right),\;I_{\mathrm{hist}}\left(x,y\right),\;I_{\mathrm{edge}}\left(x,y\right)\right]$$

This channel fusion enriches the input feature space, providing the model with intensity, contrast-enhanced structures, and fine-grained edge boundaries.

### 3.3. Data Transformation and Augmentation

To improve model generalization and minimize overfitting, the proposed framework incorporates on-the-fly data augmentation applied to preprocessed handwriting images before classification. Augmentation is performed using Keras’ ImageDataGenerator, which randomly applies transformations while retaining label integrity. Equation (Equation 5) shows that the images are rotated within ±15°, introducing orientation invariance:(5)$$I^{\prime }\left(x,y\right)=I\left(x\cos\theta -y\sin\theta ,\;x\sin\theta +y\cos\theta \right)$$

Images are shifted horizontally or vertically by a fraction of the total width or height, as shown in Equation (Equation 6):(6)$$I^{\prime }\left(x,y\right)=I\left(x-\Delta x,\;y+\Delta y\right)$$

A shear matrix induces slanting effects that mimic distortions in handwriting strokes, as shown in Equation (Equation 7):(7)$$\left[\begin{array}{c} x^{\prime }\\ y^{\prime } \end{array}\right]=\left[\begin{array}{cc} 1 & \lambda \\ 0 & 1 \end{array}\right]\left[\begin{array}{c} x\\ y \end{array}\right]$$

Equation (Equation 8) shows that random zoom operations replicate differences in handwriting scale across subjects:(8)$$I^{\prime }\left(x,y\right)=I\left(sx,\;sy\right),\;s>0$$

Images are mirrored along the vertical axis to account for left–right drawing differences. By combining augmentation with preprocessing, the framework expands the effective dataset while preserving relevant handwriting traits such as tremor-induced irregularities, stroke discontinuities, and distortions.

### 3.4. Model Architecture

To provide a comparative baseline, a standard CNN was also trained individually on the spiral and wave handwriting datasets. This allowed assessment of performance differences between a conventional CNN and the proposed approach. The proposed classification framework adopts the EfficientNetV2-S architecture as its backbone, selected for its balance of efficiency, scalability, and robust performance in image-based recognition tasks. EfficientNetV2-S utilizes fused MBConv layers in the initial stages, depthwise separable convolutions, and the SiLU activation function, enabling the model to capture fine-grained spatial features while optimizing computational cost.

Unlike conventional natural images, this study uses synthetic RGB handwriting inputs generated through a custom preprocessing pipeline. Input images are normalized using transforms.Normalize (mean = [0.5, 0.5, 0.5], std = [0.5, 0.5, 0.5]) to scale pixel values to the range [−1,1] for stable training. Each color channel encodes distinct yet complementary information: grayscale intensity for overall stroke structure, histogram-equalized contrast to enhance visibility of fine details, and Canny edge maps to highlight contour variations. This composite representation enriches the feature space, allowing the model to detect stroke irregularities and motor impairments indicative of PD. The EfficientNetV2-S model is pretrained on ImageNet and fine-tuned for binary classification. Its convolutional blocks operate as shown in Equation (Equation 9).(9)$$f\left(x\right)=\sigma \left(BN\left(\sum_{i}W_{i}*x_{i}\right)\right)$$
where *x* represents the input feature map, $W_{i}$ the convolutional kernels, ∗ the convolution operator, BN batch normalization, and σ the SiLU activation.

To tailor the EfficientNetV2-S architecture for PD detection, the original classification head is replaced with a custom sequence of layers: a Global Average Pooling (GAP) layer to reduce spatial dimensions, followed by a fully connected dense layer with 512 units and ReLU activation to learn compact feature embeddings. A dense output layer with 2 units and softmax activation is used for binary classification: healthy and Parkinson’s. Equation (Equation 10) shows the training objective for optimization using categorical cross-entropy loss:(10)$$L=-\sum_{i=1}^{N}y_{i}\;\log\left({\hat{y}}_{i}\right)$$
where $y_{i}$ denotes the ground-truth label and ${\hat{y}}_{i}$ denotes the predicted probability distribution. Details of the hyperparameter configuration of the EfficientNetV2-S model are shown in Table 2.

### 3.5. Training Strategy

The proposed framework adopts a supervised learning paradigm, in which handwriting samples are explicitly labeled as healthy or Parkinson’s. The training strategy was carefully designed to achieve a balance between fast convergence, strong generalization to unseen data, and robustness against handwriting variability, ensuring reliable classification performance. The dataset was partitioned into training, validation, and test sets. Training samples were preprocessed into synthetic RGB representations and further diversified through on-the-fly augmentation, ensuring greater variability and reducing overfitting risks. The validation set was employed for hyperparameter tuning and learning rate adjustments, while the independent test set was reserved exclusively for final model evaluation.

Model parameters were optimized using the Adam optimizer, chosen for its adaptive learning rate adjustment mechanism that accelerates convergence. The initial learning rate was set to 0.001 and was dynamically adjusted using a ReduceLROnPlateau scheduler, which decreases the learning rate when validation loss stagnates over a predefined number of epochs. To further control overfitting, multiple regularization strategies were applied. A dropout rate of 0.5 was introduced in the dense layers, randomly deactivating neurons during training to encourage more robust feature learning. Additionally, data augmentation, including rotation, shifting, shearing, zooming, and flipping, was performed to improve generalization by exposing the model to varied transformations of handwriting samples. The network was trained for up to 10 epochs with a batch size of 16. Early stopping was employed to terminate training if validation loss failed to improve across consecutive epochs, preventing unnecessary computation and minimizing the risk of overfitting. After each epoch, the model’s performance was assessed on the validation set using accuracy, precision, recall, and F1-score. These metrics provided continuous feedback on generalization capability and guided adjustments in training dynamics.

### 3.6. Evaluation Metrics

To comprehensively evaluate the proposed handwriting-based PD detection framework, several performance metrics were used. These metrics not only assess overall classification capability but also the balance between the ability to detect Parkinson’s cases and healthy individuals.

#### 3.6.1. Accuracy

Accuracy [40] measures the proportion of correctly classified samples out of the total, as shown in Equation (Equation 11).(11)$$\mathrm{Accuracy}=\frac{TP+TN}{TP+TN+FP+FN}$$

#### 3.6.2. Precision

Precision [40] quantifies how many predicted Parkinson’s cases are actually correct, as shown in Equation (Equation 12).(12)$$\mathrm{Precision}=\frac{TP}{TP+FP}$$

#### 3.6.3. Recall

Recall [40] indicates how effectively the model detects true Parkinson’s cases, as shown in Equation (Equation 13).(13)$$\mathrm{Recall}=\frac{TP}{TP+FN}$$

#### 3.6.4. F1-Score

The F1-score [40] is the harmonic mean of precision and recall, balancing both metrics, as shown in Equation (Equation 14).(14)$$F1-\mathrm{Score}=2\times \frac{\mathrm{Precision}\times \mathrm{Recall}}{\mathrm{Precision}+\mathrm{Recall}}$$

#### 3.6.5. ROC Curve

The Receiver Operating Characteristic (ROC) curve [40] plots the true positive rate (TPR) against the false positive rate (FPR) at various thresholds. The AUC score summarizes this performance; a higher value indicates better discrimination between classes.

#### 3.6.6. Confusion Matrix

The confusion matrix [40] provides a tabular view of predictions versus actual labels, allowing detailed analysis of correct classifications and misclassification patterns between healthy and Parkinson’s categories.

## 4. Experimental Results and Discussion

This section presents the experimental results derived from training and evaluating the proposed EfficientNetV2-S-based framework on the Parkinson’s Augmented Handwriting Dataset. The outcomes are systematically analyzed in terms of classification performance, robustness across handwriting variations, and comparisons with existing state-of-the-art approaches.

### 4.1. Experimental Setup

All experiments were conducted in a Python 3.11.13 environment using the TensorFlow–Keras deep learning framework. The key libraries and their versions were PyTorch 2.8.0, Torchvision 0.23.0, OpenCV 4.12.0, NumPy 2.0.2, and Scikit-learn 1.6.1. For Canny edge detection, a central preprocessing step, OpenCV’s default parameters were used with automatic threshold calculation. This precise software environment ensures full reproducibility of our preprocessing and training pipeline. Model development and training were carried out in Google Colab Pro, which provides access to an NVIDIA Tesla T4 GPU with 15 GB of RAM and 100 GB of cloud storage.

The implementation leveraged pretrained ImageNet weights for EfficientNetV2-S, with fine-tuning performed on the handwriting dataset. Training was configured with a batch size of 16 for a maximum of 10 epochs, employing early stopping to terminate training when the validation loss failed to improve. Optimization was performed using the Adam optimizer with an initial learning rate of 0.001, dynamically adjusted via a ReduceLROnPlateau scheduler. To ensure reproducibility, random seeds were fixed for dataset shuffling and parameter initialization. On-the-fly data augmentation was performed using Keras’ ImageDataGenerator, enabling exposure to new handwriting variations at every epoch. Model performance was evaluated using accuracy, precision, recall, F1-score, ROC curves, and confusion matrices.

### 4.2. Performance Evaluation on Spiral Handwriting

The model trained on spiral handwriting samples achieved 0.9868 accuracy, as shown in Figure 3. The validation results revealed that spiral drawings contained strong discriminative cues for detecting PD, particularly due to motor impairments, which manifested as tremor-induced oscillations, irregular stroke curvatures, and inconsistent line smoothness. These pathological features allowed the framework to reliably distinguish between healthy and Parkinson’s cases, highlighting the diagnostic relevance of spiral-based handwriting analysis. The EfficientNetV2-S model exhibited excellent classification performance on the spiral handwriting dataset. As shown in Table 3, the model obtained a precision of 1.00 and a recall of 0.97 for the healthy class, and a precision of 0.97 and a perfect recall for the Parkinson’s class, resulting in an F1-score of 0.99 for both classes. These metrics indicate a well-balanced model with minimal false positives and false negatives, demonstrating its ability to effectively distinguish Parkinson’s-affected handwriting from healthy samples.

The confusion matrix, as shown in Figure 4, revealed a low rate of misclassifications, with most errors arising in cases where early-stage Parkinson’s spirals exhibited a close resemblance to healthy patterns. This suggests that the subtle manifestations of motor impairments at the early stage pose a greater challenge for accurate classification.

Figure 5 shows the ROC curve of the proposed EfficientNetV2-S model on the spiral dataset.

### 4.3. Performance Evaluation on Wave Handwriting

Figure 6 shows that wave handwriting samples also proved effective in distinguishing between Parkinson’s and healthy images, achieving 98.10% accuracy. Unlike spirals, wave patterns emphasize handwriting fluidity and rhythmic consistency, thereby revealing impairments in motor smoothness.

Table 4 presents the classification results of the proposed EfficientNetV2-S model on the wave handwriting dataset. It achieved a precision of 0.95 and a recall of 1.00 for the healthy class, while for the Parkinson’s class, it attained perfect precision (1.00) and a recall of 0.97, resulting in F1-scores of 0.97 and 0.99, respectively. These results indicate that the model maintained sensitivity and specificity across both classes, effectively minimizing both false negatives and false positives, and further validating its robustness for PD detection using wave handwriting tasks.

The confusion matrix, as shown in Figure 7, revealed that most errors occurred in borderline cases where subtle motor irregularities in healthy-like waves overlapped with early Parkinson’s manifestations.

The model achieved high accuracy, precision, recall, and F1-score, further supported by a strong AUC value, indicating reliable performance, as illustrated in Figure 8.

### 4.4. Cross-Validation Performance

To further validate the robustness of the proposed framework, 5-fold cross-validation was performed on the combined dataset. The dataset was partitioned randomly into five equal subsets, with each fold serving as the validation set once, while the remaining four folds were used for training. This same process was applied across all folds, and the average performance is reported to ensure fair evaluation. The cross-validation results indicated strong generalization ability with minimal performance variance across folds. The accuracy per fold is illustrated in Figure 9.

The model demonstrated consistently high validation accuracy across all folds, ranging from 96.28% to 99.47%, reflecting its strong ability to generalize. Fold 4 showed the weakest performance, with a slightly higher validation loss (0.0968) and lower accuracy (96.28%), indicating some variability in that subset of data. In contrast, Fold 5 achieved the best results, delivering a near-perfect accuracy of 99.47% and the lowest validation loss (0.0342), highlighting excellent discriminative capability. Overall, the validation losses remained low across all folds, underscoring the model’s stability and robustness. Table 5 shows a summary of the 5-fold validation accuracy, along with the standard deviation, demonstrating the performance of the proposed model.

These results confirm that the EfficientNetV2-S-based framework is not only highly accurate but also stable across different data partitions, thereby mitigating the risk of overfitting or bias toward a specific training–validation split.

### 4.5. Baseline CNN Performance

To establish a benchmark, a standard CNN was trained using the same preprocessed spiral and wave handwriting datasets. The CNN achieved validation accuracies of 0.7143 on wave images and 0.7632 on spiral images, as shown in Figure 10. While these results indicate that the CNN captured relevant handwriting features for Parkinson’s detection, its performance was substantially lower compared to the proposed EfficientNetV2-S framework. These findings highlight the advantage of deeper architectures with transfer learning and enhanced preprocessing in capturing subtle neuromotor irregularities.

The baseline CNN model showed noticeable variations in performance across both handwriting tasks. For the spiral dataset, the model achieved a high recall of 0.95 but a lower precision of 0.69 for the healthy class, resulting in an F1-score of 0.80. Conversely, for the Parkinson’s class, the model recorded a higher precision of 0.92 but a significantly lower recall of 0.58, leading to an F1-score of 0.71. For the wave dataset, performance declined further, with the model obtaining precision, recall, and F1-score values of 0.61, 0.58, and 0.59, respectively, for the healthy class. The model achieved slightly better results for the Parkinson’s class, with a precision of 0.77, a recall of 0.79, and an F1-score of 0.78. These results indicate that the CNN baseline struggled with balanced detection, particularly in distinguishing healthy samples, reinforcing the need for advanced models to improve sensitivity and robustness across both handwriting tasks. The classification results of the standard CNN model on spiral and wave handwriting tasks are shown in Table 6.

Figure 11 presents confusion matrices of the baseline CNN model on both the spiral and wave datasets, highlighting class-wise prediction strengths and misclassifications.

Figure 12 shows the ROC curves of the baseline CNN model, illustrating its ability to distinguish between healthy and Parkinson’s classes on both the spiral and wave datasets.

### 4.6. Comparison of Performance of CNN Baseline and EfficientNetV2-S Models

To assess the effectiveness of the proposed approach, we compared its performance against a standard CNN baseline on both the spiral and wave handwriting datasets. Figure 13 summarizes the key evaluation metrics. For spiral handwriting, EfficientNetV2-S achieved an accuracy of 98.68%, precision of 0.97, recall of 1.00, and an F1-score of 0.99, demonstrating almost perfect classification. The CNN baseline achieved only 76.32% accuracy, with a lower recall of 0.58 and an F1-score of 0.71, indicating misclassifications. For wave handwriting, EfficientNetV2-S again outperformed the baseline, achieving an accuracy of 98.10%, precision of 1.00, recall of 0.97, and an F1-score of 0.99. The CNN baseline achieved an accuracy of 71.43%, precision of 0.71, recall of 0.79, and an F1-score of 0.78, showing moderate ability to identify PD cases but with reduced consistency. These results highlight that while CNNs can identify handwriting patterns, their ability to capture subtle neuromotor irregularities is limited. In contrast, the EfficientNetV2-S model, combined with preprocessing and augmentation strategies, delivers consistently superior and more reliable performance across both handwriting tasks.

### 4.7. External Validation on an Independent Dataset

To assess the generalizability and robustness of our trained models, we performed external validation using the independent Handwritten Parkinson’s Disease Augmented Dataset [41]. Our pre-trained EfficientNetV2-S models for spiral and wave tasks were applied to this new dataset without any further fine-tuning, after processing the images through our standardized preprocessing pipeline. The models demonstrated strong performance on this unseen data, accurately predicting labels for random samples. Representative examples of correctly classified handwriting samples from the external validation dataset are illustrated in Figure 14, showcasing the model’s ability to accurately recognize both spiral and wave patterns associated with Parkinson’s disease. This successful external validation confirms that the discriminative features learned by our framework are robust and transferable beyond the original training dataset.

### 4.8. Discussion

This study presented a handwriting-based framework for PD detection, integrating spiral and wave handwriting tasks, advanced preprocessing techniques, and fine-tuning of the EfficientNetV2-S architecture. The strong predictive performance and robustness of the proposed framework validate its effectiveness. A preprocessing pipeline involving histogram equalization, Canny edge detection, and synthetic RGB fusion improved feature extraction compared to using raw grayscale images. The results confirm this assertion. The preprocessing pipeline enhanced contrast and highlighted stroke boundaries, enabling the network to learn subtle motor irregularities more effectively. This demonstrates that preprocessing is a critical factor in improving generalization and sensitivity.

Individually, the models trained on spiral and wave inputs achieved strong accuracies of 98.68% and 98.10%, respectively. However, when both modalities were fused, a mean validation accuracy of 0.9767 ± 0.0109 was achieved across five folds. These results suggest that spiral drawings capture tremor-induced distortions and curvature irregularities, while wave patterns exhibit motor rhythm and smoothness. Together, they create a richer representation of motor function. The fine-tuning of EfficientNetV2-S on handwriting data yielded higher accuracy and robustness compared to conventional CNNs and handcrafted feature-based approaches. Experimental evidence supports this claim. EfficientNetV2-S consistently delivered high accuracy across folds. Its fused MBConv blocks, SiLU activation, and progressive learning strategy effectively leveraged limited handwriting data while maintaining computational efficiency, making it well-suited for clinical and real-world applications.

The baseline CNN results on the wave and spiral datasets, with accuracies of 71.43% and 76.32%, respectively, provide further evidence that simple convolutional architectures can identify gross handwriting differences between healthy and Parkinson’s subjects. The superior performance of EfficientNetV2-S underscores the importance of deeper feature hierarchies and robust preprocessing pipelines for reliable clinical applications. The proposed approach demonstrates that handwriting analysis combined with modern deep learning techniques can achieve near state-of-the-art diagnostic performance. The key contributions include establishing a reliable preprocessing pipeline to enrich feature representation from simple handwriting samples, demonstrating that combining spiral and wave tasks enhances diagnostic accuracy and stability, and validating EfficientNetV2-S as an efficient yet powerful backbone for handwriting-based biomedical analysis.

Earlier studies primarily relied on handcrafted feature extraction, such as stroke length, curvature, and velocity, combined with classifiers including support vector machines (SVMs) and random forests. Although computationally efficient, these methods exhibited limited generalization capacity and strong dependence on manual feature design. Subsequent research explored applying CNNs directly to spiral or wave handwriting samples. While CNNs successfully captured local pixel-level features, they often failed to generalize across diverse handwriting styles and required substantial preprocessing to mitigate inter-subject variability.

In contrast, the proposed EfficientNetV2-S-based framework integrates a structured preprocessing pipeline incorporating grayscale conversion, histogram equalization, and edge detection, followed by synthetic RGB fusion, enabling the model to learn richer and more discriminative representations of handwriting irregularities. The incorporation of data augmentation further enhances robustness, while transfer learning from ImageNet weights enables efficient convergence despite the limited size of the medical dataset. The key advantages of the proposed approach are the elimination of manual feature engineering, as stroke-level irregularities are learned directly from enriched images. Stable performance across folds (±0.0109) demonstrated that the proposed model outperformed earlier CNN-based studies that exhibited greater variability due to dataset imbalance and smaller sample sizes. Balanced accuracy and computational efficiency position EfficientNetV2-S as a practical solution for deployment in real-world clinical applications.

Despite promising results, several limitations remain. The risk of overfitting remains non-trivial given the modest underlying dataset, despite our efforts to mitigate it through extensive data augmentation and cross-validation. Although these strategies are effective, they may not fully capture the immense biological and stylistic variability present in the general population. This leads directly to the significant challenge of real-world deployment. The model’s performance could be substantially impacted by handwriting variations influenced by factors completely outside the current dataset’s scope, such as diverse literacy levels, cultural writing conventions, and different digital acquisition hardware. Therefore, while the augmentation strategy improves robustness, it does not eliminate the fundamental need for larger and more demographically heterogeneous datasets to ensure true generalizability.

However, the successful external validation of our framework on an independent dataset provides strong preliminary evidence of its generalizability. The model’s ability to accurately classify samples from an unseen dataset suggests that the learned feature representations of Parkinson’s spiral and wave handwriting tasks are robust and transferable, mitigating the immediate concerns of overfitting to the original dataset’s specific characteristics. The current approach focuses on static handwriting images. Incorporating temporal dynamics such as stroke velocity, pen pressure, and trajectory could provide richer diagnostic cues. Practical deployment will require validation through clinical trials, explainable artificial intelligence (XAI) tools, and user-friendly interfaces to foster trust among healthcare professionals. Addressing the critical need for trust and interpretability, highlighted through the integration of XAI techniques such as Gradient-weighted Class Activation Mapping (Grad-CAM), could be pursued in subsequent work. These methods would generate visual explanations by highlighting the specific regions in a spiral or wave drawing that most influenced the model’s classification decision. For instance, we would expect a trustworthy model to focus on areas exhibiting tremor, micrographia, or irregular stroke curvature for Parkinson’s disease samples, while highlighting smooth, continuous trajectories for healthy controls. The development and validation of such explainable interfaces are necessary steps for the clinical adoption of this technology.

### 4.9. State-of-the-Art Comparison

The proposed framework was compared with several recent studies on handwriting-based PD detection, encompassing both traditional machine learning and deep learning approaches. Comparative analysis, as shown in Table 7, highlights the key advantages of the proposed approach through the elimination of manual feature engineering, as stroke-level irregularities are learned directly from enriched images. Stable performance across folds (0.9767 ± 0.0109) demonstrates that the proposed model outperforms earlier CNN-based studies that exhibited greater variability due to dataset imbalance and smaller sample sizes.

## 5. Conclusions

This study proposed a robust handwriting-based framework for PD detection, integrating spiral and wave handwriting samples, a comprehensive preprocessing pipeline, and fine-tuning of the EfficientNetV2-S architecture. The use of grayscale normalization, histogram equalization, and edge detection, fused into a synthetic RGB representation, improved the visibility of subtle handwriting irregularities. This enriched feature space allowed deeper networks to better identify early-stage motor impairments. The model achieved accuracies of 98.68% on the spiral dataset and 98.10% on the wave dataset. By combining spiral and wave tasks, the framework achieved strong predictive performance, with a mean validation accuracy of 0.9767 ± 0.0109 across five folds, confirming its stability and generalizability. In addition to the proposed framework, a standard CNN baseline was implemented for comparative purposes. The CNN achieved validation accuracies of 0.7143 on wave handwriting and 0.7632 on spiral handwriting, substantially lower than those of the proposed EfficientNetV2-S framework. This performance gap clearly highlights the advantage of integrating advanced preprocessing and transfer learning techniques for handwriting-based Parkinson’s detection. The framework captured complementary neuromotor features, with spiral drawings reflecting tremor and irregular curvature, and wave patterns exhibiting smoothness and rhythmic control. Transfer learning enabled efficient adaptation to relatively small domain-specific datasets. The model maintained computational efficiency and outperformed traditional handcrafted features and baseline CNN architectures.

Future work will explore the integration of handwriting data with other biomarkers such as voice analysis (for detecting hypophonia and monotonicity), gait analysis (to assess stride variability and postural instability), and data from wearable sensors (to monitor tremor and bradykinesia during daily activities). The integration of a multimodal diagnostic system may be implemented through late-fusion strategies, combining likelihood scores from handwriting, voice, and gait models using a meta-classifier such as a random forest or a simple neural network to produce a final, more reliable prediction. Alternatively, a unified deep learning model could be developed to process these diverse data streams simultaneously, using dedicated subnetworks for each modality and merging their features at a later stage. This design would harness the complementary strengths of different biomarkers: handwriting for fine motor control, voice for vocal motor function, and gait for large-scale motor coordination, thereby enabling a more comprehensive and accurate digital assessment framework for PD. The deployment of lightweight, edge-ready models for mobile or telemedicine platforms will also be pursued to enable scalable, low-cost neurological screening.

## Figures and Tables

**Figure 1 diagnostics-15-02795-f001:**
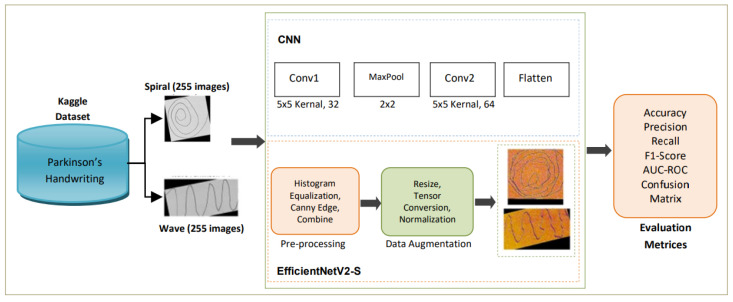
Methodology diagram of the proposed study.

**Figure 2 diagnostics-15-02795-f002:**
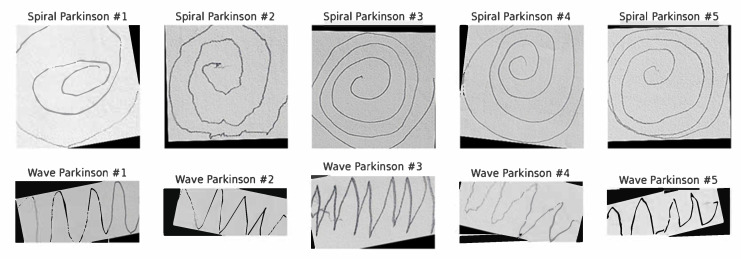
Samples of spiral and wave Parkinson’s images.

**Figure 3 diagnostics-15-02795-f003:**
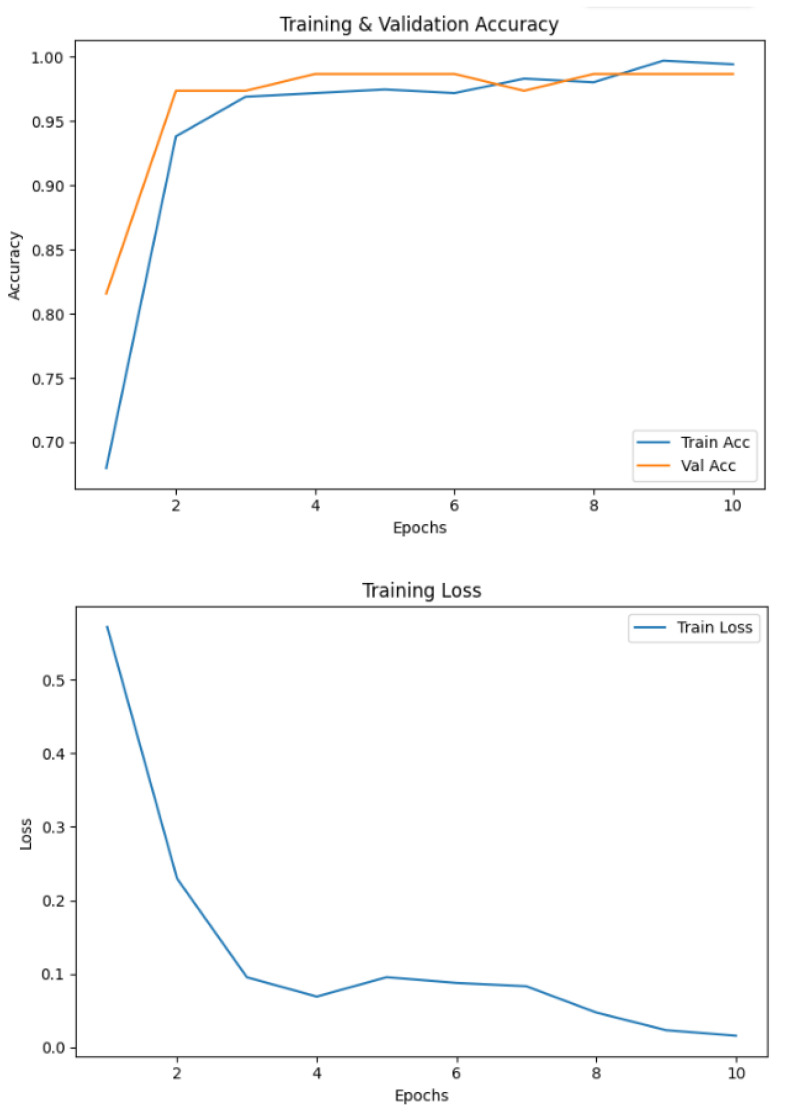
Accuracy and loss of the EfficientNetV2-S model on the spiral dataset.

**Figure 4 diagnostics-15-02795-f004:**
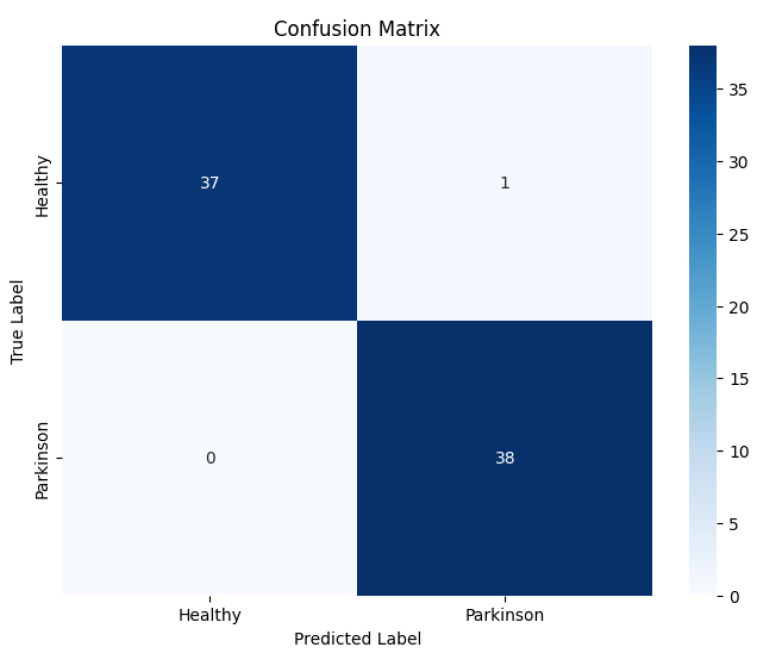
Confusion matrix of the proposed EfficientNetV2-S model on the spiral dataset.

**Figure 5 diagnostics-15-02795-f005:**
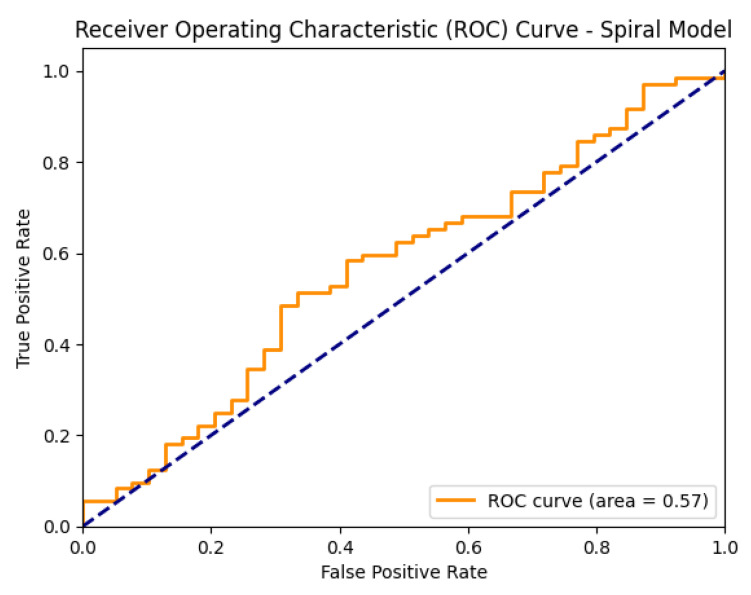
ROC curve of the proposed EfficientNetV2-S model on the spiral dataset.

**Figure 6 diagnostics-15-02795-f006:**
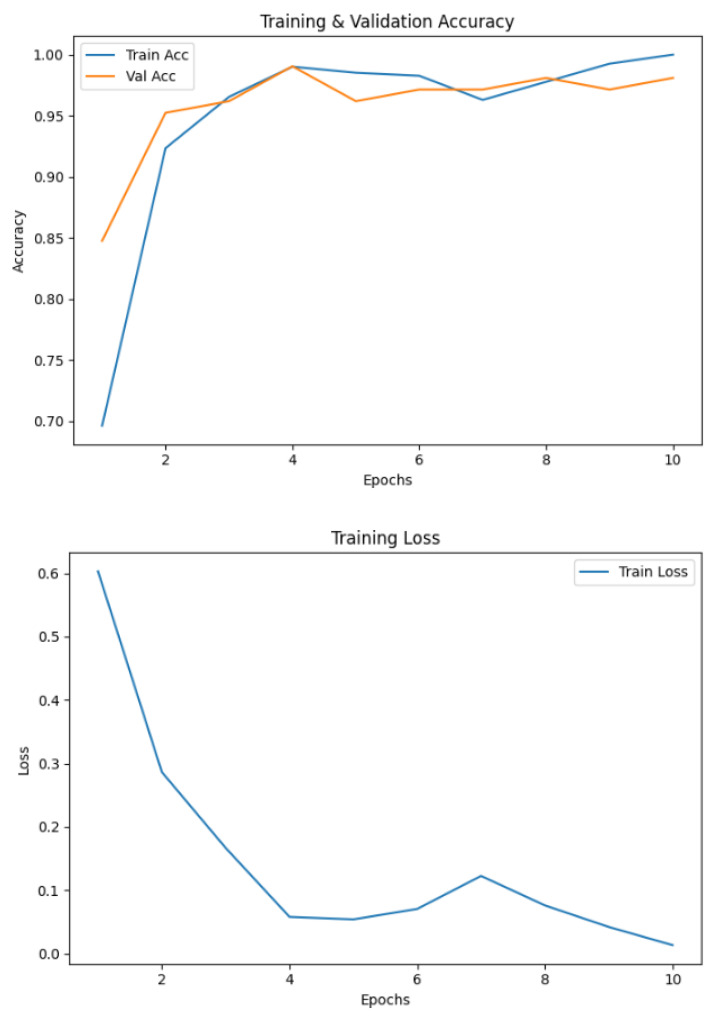
Accuracy and loss graphs of the proposed model on the wave dataset.

**Figure 7 diagnostics-15-02795-f007:**
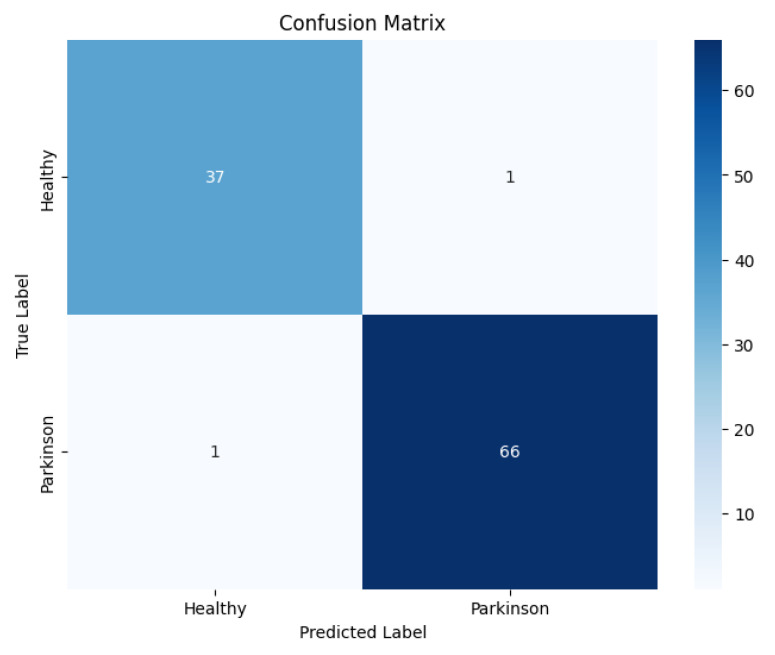
Confusion matrix of the proposed model on the wave dataset.

**Figure 8 diagnostics-15-02795-f008:**
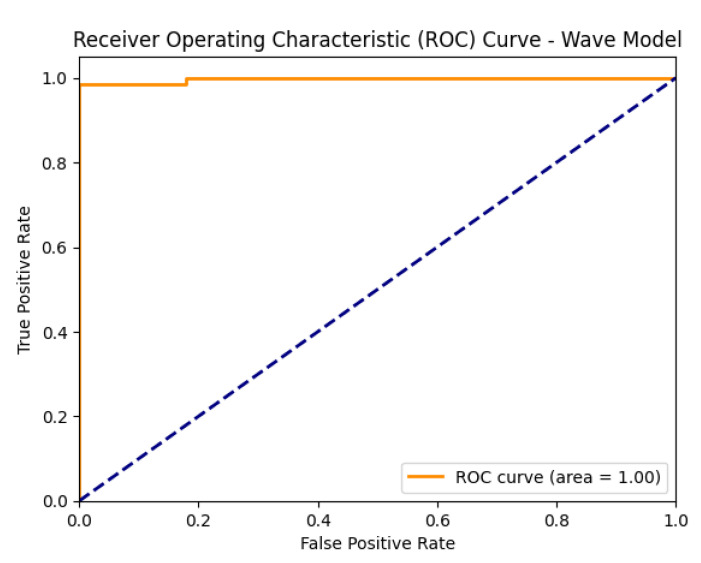
ROC curve of the proposed model on the wave dataset.

**Figure 9 diagnostics-15-02795-f009:**
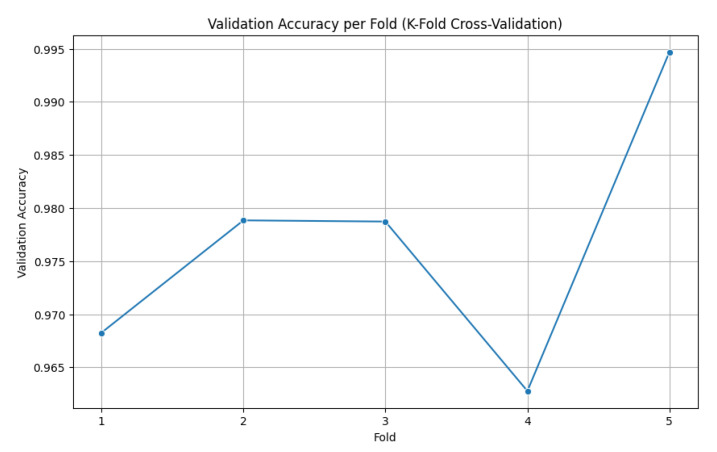
Accuracy per fold (5-fold cross-validation).

**Figure 10 diagnostics-15-02795-f010:**
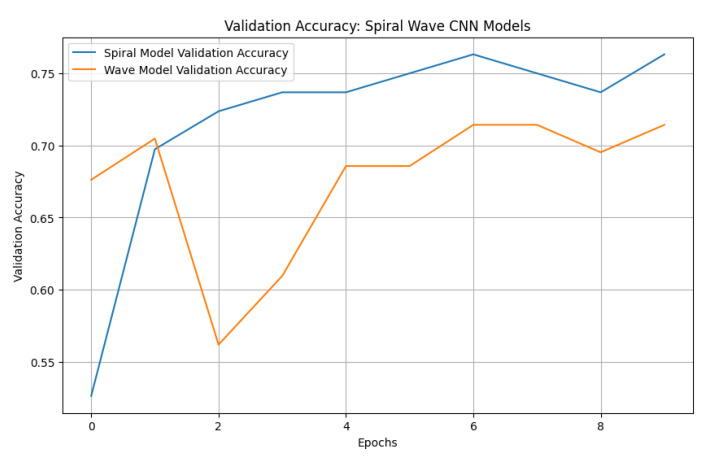
Validation accuracies of spiral and wave CNN models.

**Figure 11 diagnostics-15-02795-f011:**
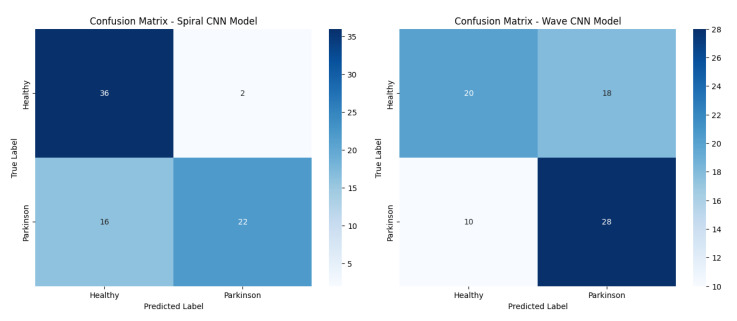
Confusion matrices of the baseline CNN models (spiral and wave).

**Figure 12 diagnostics-15-02795-f012:**
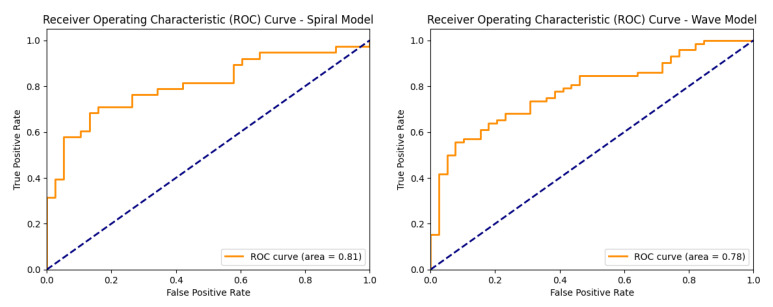
ROC curves of the baseline CNN models (spiral and wave).

**Figure 13 diagnostics-15-02795-f013:**
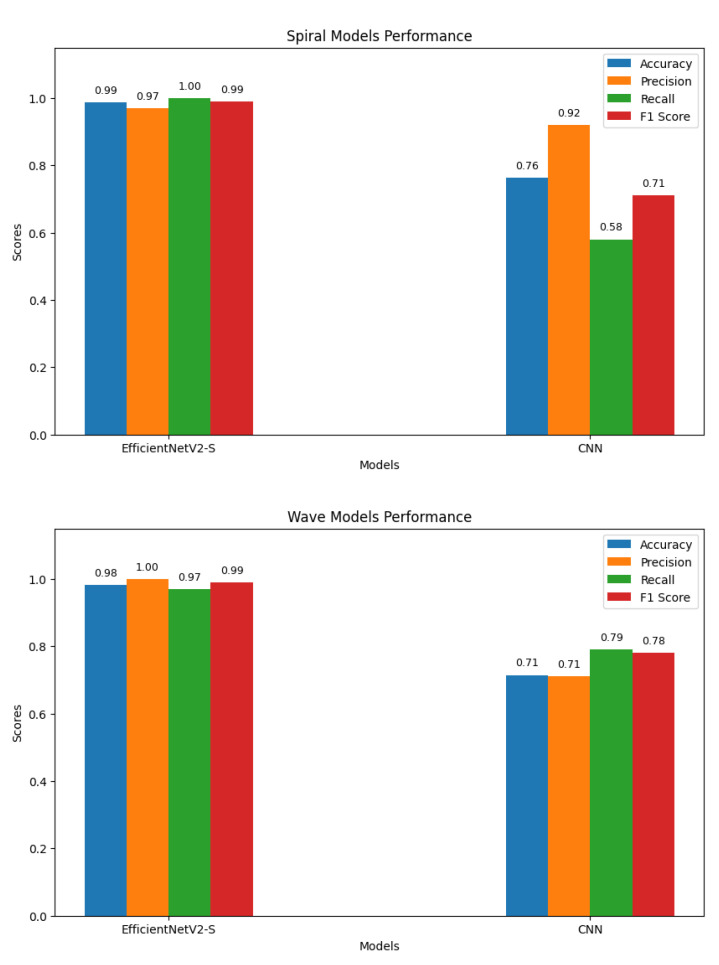
Graphs of performance metrics for the EfficientNetV2-S and CNN models.

**Figure 14 diagnostics-15-02795-f014:**
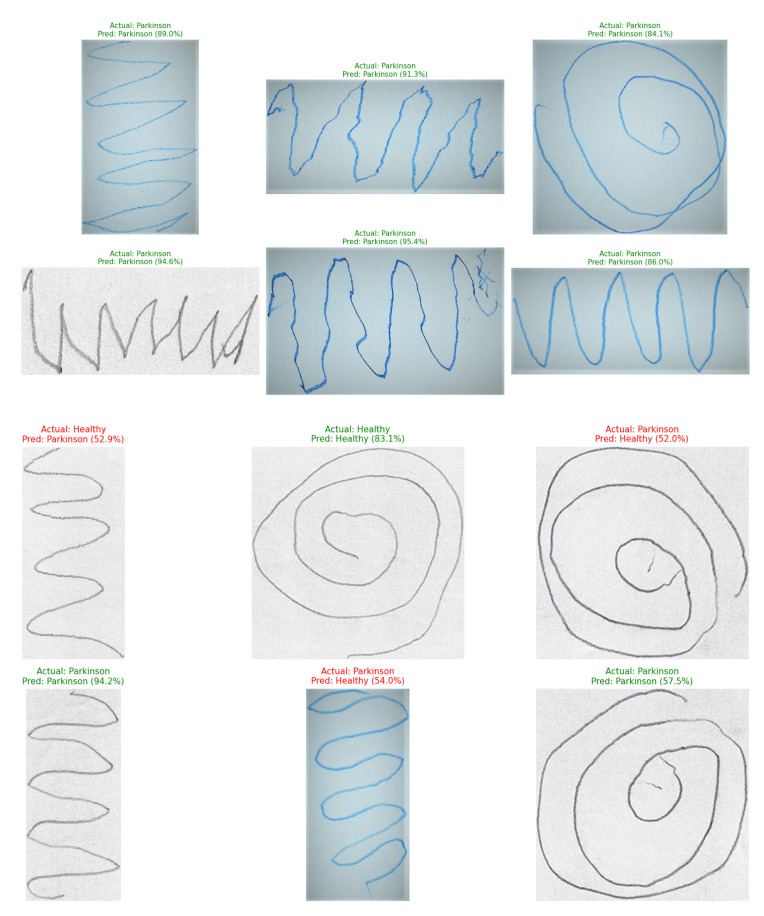
Prediction results of the proposed EfficientNetV2-S model on Parkinson’s spiral and wave handwriting samples from the external dataset.

**Table 1 diagnostics-15-02795-t001:** Summary of related works on PD detection using handwriting analysis.

Ref.	Year	Dataset	Methodology	Key Focus	Results	Limitation
[27]	2018	206 specimens from 62 subjects (31 PD, 31 controls)	Dynamic handwriting analysis	Spiral sketching features for PD detection	0.933 AUC	Small dataset, limited to spiral tasks
[28]	2020	Handwriting data from 55 PD patients	Dual CNNs on spiral and wave sketches with ensemble voting	Handwriting-based PD detection	93.3% accuracy	Small dataset, limited generalizability
[29]	2022	Handwriting visual samples	Modified SqueezeNet deep learning model	Handwriting visual features for PD prediction	90% accuracy	Lacks details on dataset size and generalizability
[25]	2023	HandPD	CAS Transformer with CycleGAN augmentation	Handwriting feature extraction for PD detection	92.68% accuracy	Limited to specific handwriting tasks
[30]	2023	4500+ augmented spiral and wave images	Inception V3, Xception	Early PD detection via hand-drawn patterns	97% accuracy	May require further validation on diverse populations and real-world settings
[32]	2024	24 PD + 24 HC handwriting samples (sentence sequences)	Domain-feature extraction, exhaustive feature selection, and SVM classifier	Capturing temporal evolution of handwriting over multiple sentences	91.67% accuracy	Relatively small sample size, relies on hand-engineered features, may be less flexible than deep learning approaches
[31]	2024	Spiral and wave drawings from PD patients and HCs	CNN-based feature extraction	Handwriting patterns for early PD detection	91.33% accuracy	Limited to spiral and wave tasks, may require larger, more diverse datasets
[33]	2024	PaHaW dataset and a new online Arabic handwriting dataset	BLSTM with Beta-stroke segmentation and Beta-elliptical feature extraction	Online handwriting patterns for PD detection	80–96% accuracy	Limited to specific handwriting styles and languages
[21]	2024	NIATS handwriting dataset (spiral + wave)	Deep transfer learning with data augmentation (AugMix, PixMix), cosine annealing	Evaluating deep learning models for spiral and wave PD detection	VGG19 model achieved 96.67% accuracy	Small dataset, limited generalization beyond that dataset
[34]	2024	Augmented spiral and wave dataset	Pre-trained VGG19 with attention mechanism	Early PD detection emphasizing critical handwriting features	97.5% accuracy	Potential generalization challenges
[35]	2025	Original handwriting dataset, augmented with edge-filtered versions (Canny, Hessian)	Compared logistic regression, decision tree, random forest, and SVM classifiers on original, edge-filtered, and augmented datasets	Evaluate whether edge detection preprocessing enhances PD risk prediction	-	-
[36]	2025	Parkinson’s Drawings Dataset	Hybrid model (CNN–LSTM)	Capturing spatial and temporal handwriting dynamics	87% accuracy	Moderate accuracy
[37]	2025	HandPD, PaHaW	CNN-based ACC-Net with attention mechanism	Enhances feature extraction from PD spiral and wave drawings	96.5% accuracy	-
[38]	2025	Handwriting images (spiral and general writing samples)	Hybrid ILN–GNet model (improved LinkNet, GhostNet) with modified Wiener filtering, PHOG, deep, and shape features	Automated PD detection using enhanced hybrid architecture	98% accuracy	Complex model architecture, preprocessing sensitive to tuning, potential overfitting

**Table 2 diagnostics-15-02795-t002:** Hyperparameter configuration of the EfficientNetV2-S model.

Hyperparameter	Value/Setting	Description
Input Image Size	224 × 224 × 3	Synthetic RGB handwriting inputs composed of grayscale, histogram-equalized, and edge-detected channels.
Batch Size	16	Number of images processed in each training step.
Optimizer	Adam	Adaptive Moment Estimation for stable and efficient convergence.
Initial Learning Rate	0.001	Starting learning rate for weight updates.
Learning Rate Scheduler	ReduceLROnPlateau	Automatically reduces learning rate when validation loss plateaus.
Loss Function	Categorical cross-entropy	Objective function for binary classification tasks.
Epochs	10	Maximum number of complete passes over the training data.
Dropout Rate	0.5	Applied in the dense layer to mitigate overfitting.
Weight Initialization	ImageNet pretrained weights	Transfer learning initialization to accelerate convergence.
Data Augmentation	Rotation, shift, shear, zoom, horizontal flip	Applied during training to enhance data diversity and generalization.
Validation Split	15%	Fraction of training data reserved for validation.
Evaluation Metrics	Accuracy, precision, recall, F1-score, AUC	Comprehensive measures for evaluating classification performance.

**Table 3 diagnostics-15-02795-t003:** Classification results of the EfficientNetV2-S model on the spiral dataset.

Class	Precision	Recall	F1-Score
Healthy	1.00	0.97	0.99
Parkinson’s	0.97	1.00	0.99

**Table 4 diagnostics-15-02795-t004:** Classification results of the EfficientNetV2-S model on the wave dataset.

Class	Precision	Recall	F1-Score
Healthy	0.95	1.00	0.97
Parkinson’s	1.00	0.97	0.99

**Table 5 diagnostics-15-02795-t005:** Fivefold cross-validation results of the proposed EfficientNetV2-S model.

Fold	Accuracy	Loss
Fold 1	0.9683	0.0732
Fold 2	0.9788	0.0492
Fold 3	0.9787	0.0468
Fold 4	0.9628	0.0968
Fold 5	0.9947	0.0342
Standard Deviation	±0.0109	

**Table 6 diagnostics-15-02795-t006:** Classification results of the baseline CNN model on the spiral and wave datasets.

	Spiral			Wave		
**Class**	**Precision**	**Recall**	**F1-Score**	**Precision**	**Recall**	**F1-Score**
Healthy	0.69	0.95	0.80	0.61	0.58	0.59
Parkinson’s	0.92	0.58	0.71	0.77	0.79	0.78

**Table 7 diagnostics-15-02795-t007:** Comparison of the proposed EfficientNetV2-S model with related works.

Ref.	Year	Model	Accuracy
[42]	2021	1D convolutions and BiGRUs	94.44%
[29]	2022	Modified SqueezeNet deep learning model	90%
[43]	2022	Random forest	95%
[30]	2023	Inception V3 and Xception	97%
[34]	2024	VGG19 with attention mechanism	97.5%
[31]	2024	VGG-19 CNN-based feature extraction	91.33%
[44]	2024	InceptionV3	89%
[36]	2025	Hybrid model (CNN–LSTM)	87%
[37]	2025	ACC-Net with attention mechanism	96.5%
Proposed study	2025	EfficientNetV2-S-based framework	98.68%

## Data Availability

The datasets used and/or analyzed during the current study are publicly available on the following URLs: https://www.kaggle.com/datasets/sowmyabarla/parkinsons-augmented-handwriting-dataset and https://www.kaggle.com/datasets/banilkumar20phd7071/handwritten-parkinsons-disease-augmented-data, accessed on 1 September 2025.

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
