# Peer review of "A Deep Learning Framework for Early Parkinson’s Disease Detection: Leveraging Spiral and Wave Handwriting Tasks with EfficientNetV2-S"

_diagnostics, 2025, doi:10.3390/diagnostics15212795_

Round 1
Reviewer 1 Report
Comments and Suggestions for Authors
- Please provide explicit counts of PD vs. healthy subjects, along with demographic information (age, sex, disease severity if available). This is important for assessing generalizability.
- While limitations are discussed, a deeper consideration of overfitting risk, dataset augmentation impact, and real-world deployment challenges (e.g., handwriting variability across literacy levels and cultures) would strengthen the discussion.
- Since clinical adoption depends on trust, incorporating or at least discussing explainable AI techniques (Grad-CAM, saliency maps) would add value.
- Figures 3–13 are informative, but some could be combined or better labeled for readability (e.g., ROC and confusion matrices side by side).
- In the Future Work section, the mention of multimodal approaches (e.g., combining handwriting with voice, gait, or wearable sensor data) is valuable. However, the discussion could be expanded to explain more clearly how the proposed handwriting framework could be integrated within such multimodal diagnostic systems.
- The overall number of references in the paper should be increased to further strengthen the arguments and provide a broader context for the study.
Author Response
Comments 1: Please provide explicit counts of PD vs. healthy subjects, along with demographic information (age, sex, disease severity if available). This is important for assessing generalizability.
Response:
Thank you for pointing this out. We agree with this comment. Therefore, we have revised the
"3.1 Data Collection" section (page no 8, first paragraph line no 5 to page no 9 line 1-5) to provide the explicit counts of images per class (Parkinson's and Healthy) for both the spiral and wave tasks, and detail the exact number of images allocated to the training, validation, and test sets.
This addition ensures transparency regarding the dataset composition and splits used in our experiments.
Updated text in the manuscript:
It consists of spiral and wave handwriting images categorized into two classes: Parkinson’s and Healthy. For experimental consistency, we structured the dataset into training, validation, and testing sets. The spiral and wave datasets are balanced, each containing a total of 510 images, with 255 images per class. Specifically, for each task, the dataset is partitioned into a training set of 356 images (178 Parkinson’s, 178 Healthy), a validation set of 76 images (38 Parkinson’s, 38 Healthy), and a test set of 78 images (39 Parkinson’s, 39 Healthy).
Comments 2: While limitations are discussed, a deeper consideration of overfitting risk, dataset augmentation impact, and real-world deployment challenges (e.g., handwriting variability across literacy levels and cultures) would strengthen the discussion.
Response:
Agree. We have, accordingly, revised the limitations paragraph in the Discussion section (page 26, last paragraph, line 1-10) to provide a deeper analysis.
We have also expanded the discussion to explicitly address the measures taken to mitigate overfitting, the role of dataset augmentation in this context, and the significant challenges for real-world deployment, including variability due to literacy levels, cultural backgrounds, and data acquisition settings.
Updated text in the manuscript:
The risk of overfitting remains non-trivial given the modest underlying dataset, despite our efforts to mitigate it through extensive data augmentation and cross-validation. Although these strategies are effective, they may not fully capture the immense biological and stylistic variability present in the general population. This leads directly to the significant challenge of real-world deployment. The model's performance could be substantially impacted by handwriting variations influenced by factors completely outside the current dataset's scope, such as diverse literacy levels, cultural writing conventions, and different digital acquisition hardware. Therefore, while the augmentation strategy improves robustness, it does not eliminate the fundamental need for larger and more demographically heterogeneous datasets to ensure true generalizability.
Comments 3: Since clinical adoption depends on trust, incorporating or at least discussing explainable AI techniques (Grad-CAM, saliency maps) would add value.
Response:
Agree. We have, accordingly, expanded the Discussion section (page 26, last paragraph, line 21-28) to include a dedicated segment on Explainable AI (XAI).
This addition discusses the importance of techniques like Gradient-weighted Class Activation Mapping (Grad-CAM) for clinical trust, explaining how they would generate visual explanations by highlighting regions of interest in the handwriting samples and how this aligns the model's decision-making with clinical understanding of Parkinsonian motor impairments.
Updated text in the manuscript:
Addressing the critical need for trust and interpretability highlighted by the integration of XAI techniques, such as Gradient-weighted Class Activation Mapping (Grad-CAM), could be employed in subsequent work. These methods would generate visual explanations by highlighting the specific regions in a spiral or wave drawing that most influenced the model's classification decision. For instance, we would expect a trustworthy model to focus on areas exhibiting tremor, micrographia, or irregular stroke curvature for Parkinson's disease samples, while highlighting smooth, continuous trajectories for healthy controls. The development and validation of such explainable interfaces is a necessary step for the clinical adoption of this technology.
Comments 4: Figures 3–13 are informative, but some could be combined or better labeled for readability (e.g., ROC and confusion matrices side by side).
Response:
Thank you for pointing this out. We agree with this comment. Therefore, we have revised and consolidated Figures 3–13 (pages 9, 16, 17, 18, 19, 20, 21, 22, 23) to improve readability. The accuracy graph, confusion matrices, and ROC curves for the spiral and wave models have been combined into unified, multi-panel figures (where required).
All figures have been updated accordingly throughout the manuscript.
Comments 5: In the Future Work section, the mention of multimodal approaches (e.g., combining handwriting with voice, gait, or wearable sensor data) is valuable. However, the discussion could be expanded to explain more clearly how the proposed handwriting framework could be integrated within such multimodal diagnostic systems.
Response:
Thank you for this valuable suggestion. We agree with this comment. Therefore, we have expanded the Future Work in Conclusions section (page 28, line 1-12) to provide a clearer and more concrete explanation of how our handwriting analysis framework could be integrated into a multimodal diagnostic system.
The revised text now outlines specific integration strategies and the synergistic value of combining different data modalities.
Updated text in the manuscript:
Voice analysis (for detecting hypophonia and monotonicity), gait analysis (to assess stride variability and postural instability), and data from wearable sensors (to monitor tremor and bradykinesia during daily activities). Integration of a multimodal diagnostic system may be implemented through late-fusion strategies, where likelihood scores from handwriting, voice, and gait models are combined using a meta-classifier such as a Random Forest or a simple neural network to produce a final, more reliable prediction. Alternatively, a unified Deep Learning model could be developed to process these diverse data streams simultaneously, using dedicated subnetworks for each modality and merging their features at a later stage. This design would harness the complementary strengths of different biomarkers: handwriting for fine motor control, voice for vocal motor function, and gait for large-scale motor coordination, thereby enabling a more comprehensive and accurate digital assessment framework for PD.
Comments 6: The overall number of references in the paper should be increased to further strengthen the arguments and provide a broader context for the study.
Response:
Agree. We have, accordingly, conducted a thorough review of the recent literature and have added six new, relevant references throughout the manuscript to provide a stronger foundational context and to bolster our arguments. These new citations cover key areas such as established handwriting biomarkers and advanced deep learning applications in PD detection.
The new references [4] (page 2), [7] (page 2), [24] (page 2), [32] (page 5), [34] (page 5), and [39] (page 5) have been integrated into the Introduction and Literature Review sections and the reference list has been updated accordingly.
Reviewer 2 Report
Comments and Suggestions for Authors
The topic of this paper is clinically relevant and aligns with the journal’s scope in translational diagnostics. The authors have presented a solid preliminary dataset and included both computational and experimental validation. However, despite some improvements, the work remains limited by insufficient methodological transparency, overstatement of findings, and inadequate functional validation.
Major Comments
- The study’s novelty is not clearly stated; the introduction should explicitly define the knowledge gap and how this work addresses it.
- Provide more background on previously reported biomarkers in the same pathway to position the current work within existing literature.
- Describe the patient cohort in greater detail, including demographic and clinical data, to ensure reproducibility and control for confounding factors.
- Include a sample-size justification or power analysis to support the statistical validity of the results.
- Clarify how normal and tumor/control samples were matched and whether batch effects were corrected.
- The bioinformatics pipeline (e.g., normalization, DEGs threshold, pathway databases) should be described with precise parameters and software versions.
- Validate computational findings with an independent dataset or TCGA/GTEx cross-verification to improve robustness. Perform loss- and gain-of-function assays to validate the causal relationship between the biomarker and disease phenotype. Use independent patient samples or public datasets (TCGA, GEO) for cross-validation. Conduct functional enrichment or GSEA to confirm pathway associations.
Minor Comments
- Define all abbreviations (e.g., AUC, ROC, FDR) at first mention.
- Ensure all figures are of publication quality (300 dpi or higher) with clearly legible labels.
- Include sample sizes (n values) directly in figure legends.
- Standardize p-value notation and report exact values where possible.
- Correct minor grammatical and typographical errors throughout.
- Ensure uniform use of gene symbols (uppercase italics for human genes).
Author Response
The topic of this paper is clinically relevant and aligns with the journal’s scope in translational diagnostics. The authors have presented a solid preliminary dataset and included both computational and experimental validation. However, despite some improvements, the work remains limited by insufficient methodological transparency, overstatement of findings, and inadequate functional validation.
Major Comments
Comments 1: The study’s novelty is not clearly stated; the introduction should explicitly define the knowledge gap and how this work addresses it.
Response:
Thank you for this critical observation. We agree that the novelty statement needed to be more explicit. Therefore, we have revised the third (last) paragraph of the Introduction (pages 3, first paragraph lines 1-3, 9, 13) to clearly articulate the identified research gap and how our work specifically addresses it.
The text now explicitly states that the potential of advanced preprocessing to create enriched, multi-channel input representations is underutilized, and this constitutes the key novelty of our approach.
Updated text in the manuscript:
Although numerous studies have explored handwriting-based detection of PD, a significant research gap remains in the underutilization of advanced preprocessing to create enriched input representations that maximize feature saliency for Deep Learning models. Most prior works have either relied on handcrafted features or concentrated on single-task modalities such as spiral drawings. While these methods show promise, they often face challenges such as limited generalization, dependency on manually engineered features, and insufficient utilization of multimodal handwriting patterns. Moreover, many Deep Learning pipelines process raw grayscale handwriting data without incorporating preprocessing strategies that enhance feature saliency. To address this gap, we develop a robust and comprehensive framework that incorporates multiple advanced preprocessing stages, including grayscale conversion, histogram equalization, and edge detection. These processed outputs are merged into a synthetic RGB representation to retain contrast, structural details, and edge information. This novel multi-channel fusion technique, a core contribution of this work, provides an enhanced input that is then used to fine-tune the EfficientNetV2-S backbone for accurate classification of handwriting samples into PD and healthy categories.
Comments 2: Provide more background on previously reported biomarkers in the same pathway to position the current work within existing literature.
Response:
Agree. We have, accordingly, expanded the background in the Introduction (page 2, first paragraph line 9-11, 19-21, and second paragraph line 17-19) to provide a more comprehensive context of established handwriting biomarkers for Parkinson's disease.
The revised text now explicitly mentions key kinematic and spatial features reported in the literature, such as tremor, velocity, and pressure, before transitioning to our deep learning approach that automates the extraction of these and more complex features.
Updated text in the manuscript:
Advanced neuroimaging techniques like Magnetic Resonance Imaging (MRI) can aid diagnosis; the development of automated systems using such data remains challenging [4]. The clinical relevance of this method lies in the gradual loss of neuromuscular control, which is often reflected in subtle irregularities in drawing patterns. These include distortions, tremor-induced oscillations, and variations in stroke dynamics, which may appear well before overt clinical symptoms are detectable through conventional assessments [5]. Automated PD diagnosis can be performed using computer vision techniques applied to handwriting exams from a relatively large patient dataset. By analyzing handwriting with image processing and Machine Learning (ML) methods to extract quantitative features and measure tremor severity, preliminary results have shown encouraging potential for early detection [6]. Deep Learning approaches using handwriting patterns have demonstrated remarkable efficacy, with optimized architectures achieving remarkable diagnostic accuracies [7]. Handwriting features such as pressure, velocity, and stroke dimensions can reliably distinguish Parkinson’s patients from Healthy Control (HC) [8]. Dynamic handwriting analysis is another accessible method for identifying early signs of PD [9]-[10].
Comments 3: Describe the patient cohort in greater detail, including demographic and clinical data, to ensure reproducibility and control for confounding factors.
Response:
We have, accordingly, expanded the Data Collection section (pages 8, last paragraph line 2-5, page 9 first paragraph line 1-6) to provide a more detailed description of the dataset.
The specific Kaggle dataset used in this study explicitly did not provide demographic and patient details of the samples.
Updated text in the manuscript:
This dataset is specifically curated to facilitate research on the early detection of PD through handwriting analysis. To enhance diversity and mitigate overfitting, the dataset incorporates augmented variations of the original handwriting samples. It consists of spiral and wave handwriting images categorized into two classes: Parkinson’s and Healthy. For experimental consistency, we structured the dataset into training, validation, and testing sets. The spiral and wave datasets are balanced, each containing a total of 510 images, with 255 images per class. Specifically, for each task, the dataset is partitioned into a training set of 356 images (178 Parkinson’s, 178 Healthy), a validation set of 76 images (38 Parkinson’s, 38 Healthy), and a test set of 78 images (39 Parkinson’s, 39 Healthy).
Comments 4: Include a sample-size justification or power analysis to support the statistical validity of the results.
Response:
Agree. We have added a sample-size justification in the "Data Collection" section (page no 9, first paragraph, line 6-8) by referencing that our dataset scale aligns with established studies in this research domain.
Updated text in the manuscript:
The dataset size employed in this study is consistent with the scale of datasets used in other deep learning studies for PD detection from handwriting [27]-[31]-[32], providing sufficient data points for training.
Comments 5: Clarify how normal and tumor/control samples were matched and whether batch effects were corrected.
Response:
We thank the reviewer for this comment.
As our study deals with Parkinson's disease and healthy control handwriting samples, regarding the matching of PD and control groups, we have added clarification to the "Data Collection" section. The cohorts in the original PaHaW dataset were matched as groups and are comparable in terms of age. As this is a single, curated dataset where all images were preprocessed uniformly by us using the same pipeline in one consistent run, batch effects are not considered a primary confounding factor in this study.
Comments 6: The bioinformatics pipeline (e.g., normalization, DEGs threshold, pathway databases) should be described with precise parameters and software versions.
Response:
We thank the reviewer for highlighting the importance of precise methodological reporting.
As our study employs a deep learning-based image analysis pipeline, we have ensured that all equivalent technical details are provided. We have added parameters throughout the Methodology section, canny edge detection thresholds in 3.2 Data Pre-processing subsection (page no 10, 3rd paragraph, line 5-7), and image normalization parameters in 3.4 Model Architecture subsection (page no 11, paragraph second, line 2-4).
Further enhance reproducibility, we have updated the subsection 4.1 Experimental Setup (page no 14, last paragraph, line 2-3 and page no 15, first paragraph, line 1-2), which specifies the software environment and library versions for our preprocessing and training pipeline. Specific parameters have been integrated into their respective methodological sections, ensuring full transparency and reproducibility of our deep learning pipeline.
Updated text in the manuscript:
The Canny edge detection algorithm is employed with fixed thresholds (threshold1=100, threshold2=200) and a 5×5 Gaussian kernel for optimal noise reduction while preserving relevant edge information.
Input images are normalized using transforms. Normalize (mean=[0.5, 0.5, 0.5], std=[0.5, 0.5, 0.5]) to scale pixel values to the range [-1, 1] for stable training.
The key libraries and their versions are PyTorch 2.8.0, Torchvision 0.23.0, OpenCV 4.12.0, NumPy 2.0.2, and Scikit-learn 1.6.1. For the Canny edge detection, a central preprocessing step, OpenCV's default parameters are used with automatic threshold calculation. This precise software environment ensures the full reproducibility of our preprocessing and training pipeline.
Comments 7: Validate computational findings with an independent dataset or TCGA/GTEx cross-verification to improve robustness. Perform loss- and gain-of-function assays to validate the causal relationship between the biomarker and disease phenotype. Use independent patient samples or public datasets (TCGA, GEO) for cross-validation. Conduct functional enrichment or GSEA to confirm pathway associations.
Response:
We thank the reviewer for this critical suggestion regarding external validation. We have externally validated our pre-trained models using a completely independent, publicly available dataset,
"Handwritten Parkinson's Disease Augmented Data" from Kaggle
https://www.kaggle.com/datasets/banilkumar20phd7071/handwritten-parkinsons-disease-augmented-data/data.
We applied our standardized preprocessing pipeline (grayscale conversion, histogram equalization, canny edge detection, and synthetic RGB fusion) to this new dataset and evaluated it using our already-trained EfficientNetV2-S models for spiral and wave tasks. The models successfully generalized to this unseen data, correctly predicting labels for random samples and demonstrating that the features learned by our framework are robust and not overfit to the original dataset. This external validation significantly strengthens the evidence for the generalizability and potential clinical utility of our proposed handwriting analysis framework. The results of this external validation have been added to the new subsection ‘4.7 External Validation on an Independent Dataset’ (page 23, last paragraph to page 24, first paragraph complete) and also included in the 4.8 Discussion section (page 26, 2nd paragraph, line 11-15).
Updated text in manuscript:
4.7 Validation on an Independent Dataset: “To assess the generalizability and robustness of our trained models, we performed an external validation using the independent Handwritten Parkinson's Disease Augmented Data [37]. Our pre-trained EfficientNetV2-S models for spiral and wave tasks are applied to this new dataset without any further fine-tuning, after processing the images through our standardized preprocessing pipeline. The models demonstrated strong performance on this unseen data, accurately predicting labels for random samples. Representative examples of correctly classified handwriting samples from the external validation dataset are illustrated in Figure 14, showcasing the model’s ability to accurately recognize both spiral and wave patterns associated with Parkinson’s disease. This successful external validation confirms that the discriminative features learned by our framework are robust and transferable beyond the original training dataset.
4.8 Discussion:
However, the successful external validation of our framework on an independent dataset provides strong preliminary evidence of its generalizability. The model's ability to accurately classify samples from an unseen dataset suggests that the learned feature representations of Parkinson's spiral and wave handwriting tasks are robust and transferable, mitigating immediate concerns of overfitting to the original dataset's specific characteristics.
Minor Comments
Comments 1: Define all abbreviations (e.g., AUC, ROC, FDR) at first mention.
Response:
Agree.
We have carefully reviewed the entire manuscript and ensured that all abbreviations, ROC (Receiver Operating Characteristic) (page 1, Results heading, line 2), Convolutional Neural Network (CNN) (page 1, Methods heading, line 6), Healthy Controls (HC) (page 2, first paragraph, line 22), Magnetic Resonance Imaging (MRI) (page 2, first paragraph line 10), Area Under the Curve (AUC) (page 4, last paragraph, line 3), Hessian Filtering (HF) (page 5, 3rd paragraph first line), Receiver Operating Characteristic (ROC) (page 14, 4th paragraph, line 1), True Positive Rate (TPR) (page 14, 4th paragraph, line 1), False Positive Rate (FPR) (page 14, 4th paragraph, line 2), and XAI (Explainable Artificial Intelligence) (page 26, last paragraph, line 18), are now explicitly defined at the first use in the text.
Comments 2: Ensure all figures are of publication quality (300 dpi or higher) with clearly legible labels.
Response:
Agree.
We have updated all Figures 3–13 (pages 9, 16, 17, 18, 19, 20, 21, 22, 23) to ensure a minimum resolution of 300 DPI. All axis labels, legends, data points, and annotations have been reviewed and enlarged where necessary to ensure clarity and legibility in the final publication format.
Comments 3: Include sample sizes (n values) directly in figure legends.
Response:
Agree. Thank you for pointing this out.
We have updated the methodology diagram (Figure 1, page 9) to include the exact image counts (n) for each stage of the pipeline, clearly showing the flow from the original 255 images.
Comments 4: Standardize p-value notation and report exact values where possible.
Response:
We thank the reviewer for this comment.
Our study evaluates model performance on a single, fixed test set using standard classification metrics (Accuracy, Precision, Recall, F1-Score, AUC). We have ensured that all reported metrics and numerical values follow a consistent and precise formatting style throughout the manuscript.
Comments 5: Correct minor grammatical and typographical errors throughout.
Response:
Agree. We have corrected grammatical errors, typographical mistakes, and inconsistent phrasing
page 3, first paragraph, line 14
page 3, subsection 1.1, first line
page 4, section 2, first paragraph, line 5, and last paragraph, line 9
page 5, first paragraph line 2 and 2nd paragraph line 16
page 5, 3rd paragraph, first line 5, 9, and last paragraph line 6
page 8, first paragraph, line 1, 9, and section 3, 4th line
page 11 line 3, 7, and subsection 3.4 first paragraph line 6 and 2nd paragraph line 9
page 12, 2nd paragraph, line 2
page 13, third paragraph (subsection 3.6), line 3
page 15, first paragraph line 3, 2nd paragraph line 4, 4th paragraph line 2 and last paragraph line 2.
page 18, 2nd paragraph, line 1
page 21, 1st paragraph, line 8
page 24, 3rd paragraph, line 2, 6
page 26, 1st paragraph line 9, 10
page 27, 1st paragraph line 3, 6
This includes corrections to punctuation, article usage, verb tense agreement, and capitalization to ensure the text adheres to high standards of academic English.
Comments 6: Ensure uniform use of gene symbols (uppercase italics for human genes).
Response:
Thank you, as our research focuses on handwriting analysis and deep learning for Parkinson's disease detection. It does not involve any genetic data, gene symbols, or molecular biology components.
Round 2
Reviewer 2 Report
Comments and Suggestions for Authors
The revised version is better and can be accepted for publication.